# Characteristics of Parking Lots Located along Main Roads in Terms of Cargo Truck Requirements: A Case Study of Poland

**Igor Betkier [1],\* and Elżbieta Macioszek [2]**

1 Institute of Logistics, Faculty of Security, Logistics and Management, Military University of Technology, Gen. Sylwester Kaliski Street 2, 00-908 Warsaw, Poland
2 Department of Transport Systems, Traffic Engineering and Logistics, Faculty of Transport and Aviation Engineering, Silesian University of Technology, Akademicka Street 2A, 44-100 Gliwice, Poland
\* Correspondence: igor.betkier@wat.edu.pl

**Abstract:** The issue related to parking lots has a multidimensional nature. For the purposes of the article, the analysis covered 113 parking lots located directly next to national roads in the Mazovian Voivodeship (Poland), which is similar in terms of economic development and conditions of the linear infrastructure to other countries in Central Europe. The conducted research included analyzing the correlation of parameters in individual parking lots and those related to nearby roads (traffic volume, technical quality) and the type of area in which they occurred (population density). In addition, the presented results describe the capacity of parking in the analyzed region in terms of the number of available parking slots for heavy goods vehicles, the size of the additional operating area, and the presence of security-related elements (fencing, lighting, physical protection, monitoring). The analysis demonstrated a correlation between some parameters (type of nearby road and security $r_s = 0.35$, area size and number of slots $r_s = 0.35$) and significant differences in results between private and public parking lots (average number of slots 18 versus 34, percent of most secured parkings 45% versus 69%). The obtained innovative results can be used by public institutions and entities providing parking services to reorganize the space at parking places and to conduct more effective analyses from the point of view of future investments in infrastructure accompanying the public road network.

**Keywords:** parking lot; road transport; transport infrastructure; parking design; transport planning; freight transport; parking space; parking security; parking capacity; parking characteristics

## 1. Introduction

Parking lots constitute an integral part of the transportation infrastructure, due to the range of functions they perform. In addition, their primary purpose, which is the possibility of temporary vehicle stopping and their technical conditions and accompanying infrastructure, offers travelers many options, which include the protection of property, the possibility of resting in a dedicated place or the purchase of goods and services (including those necessary to continue the journey). A continuous growth trend in the share of road transport in the total freight work is observed worldwide [1,2]. On the other hand, this trend may change due to modern solutions for land transport, focusing, for example, on the more environmentally friendly railway. Nevertheless, the majority share of road transport is likely to remain unchanged in the future due to, among others, new optimization methods and solutions [3–5], which means that the aspects related to the accompanying infrastructure will remain relevant.

Against the background of issues related to the subject of parking lots, publications describing methods and tools for finding a free parking space in an urban area [6–9], parking management [10,11], and analysis of data about their users stand out in the literature [12,13]. A particular problem is posed by the establishment of appropriate technical parameters

for vehicle stops in large urban agglomerations, where available space is limited. As a result, multi-level solutions are proposed, based on special traffic organization, modern technologies, and requiring appropriate urban space management [14].

The issue of parking lots has a multidimensional character. In the analyzed thematic area, it is noticeable that a relatively small number of analyses have been conducted in relation to the properties of the existing parking lots. Those that have been conducted often focus on specific aspects without a broader context which would be useful for a full description of the phenomenon. Table 1 presents publications having the greatest similarity with regard to their subjects, containing an analysis of parking properties from the point of view of individual problems.

**Table 1.** Related works summary.

| Study | Analyzed Parameters | Methodology | Weaknesses |
|---|---|---|---|
| [15] | location, traffic, occupancy | Observation of parking lot (1) occupancy | No analysis related to the correlation between parameters, lack of data |
| [16] | location, space, occupancy | Observation of parking lots (3) occupancy | Parking capacity is expressed only as percent of parking space usage (in m²), lack of data |
| [17] | location, space, occupancy | GIS-based fuzzy analytical hierarchy process (AHP) approach | Problem analyzed on a micro scale |
| [18] | slots, location, occupancy | Observation of parking lots (3) occupancy | Lack of data |
| [19] | slots, space | Modelling of the movement of vehicles | Lack of parameters |
| [20] | location, space, occupancy | Observation of parking lot (1) occupancy | Lack of parameters, lack of data |
| [21] | location, occupancy, correlation | Observation of parking lots (2) occupancy | Problem analyzed on a micro scale |
| [22] | security, location | Analysis of TAPA EMEA IIS incident database | Difficult to extract a trend from a limited data set |
| [23] | slots, location, occupancy | Survey distributed within public and private sector stakeholders | No analysis related to the correlation between parameters, subjective opinions of respondents |
| [24] | slots, security, location, occupancy | Survey distributed within public and private administrations, haulage companies and truck drivers, federal highway administration (FHWA) model | Problem analyzed on a micro scale |
| [25] | location, occupancy | Observation of parking lots (2) occupancy | Lack of parameters, problem analyzed on a micro scale |
| [26] | slots, location, occupancy | Survey distributed within truck drivers | Problem analyzed on a micro scale |
| [27] | slots, security, density | Analysis of rest areas managed by national highway company | Only public parking lots analyzed, lack of results presented |
| [28] | location, occupancy | Overnight parking choice model based on traffic analysis zones (TAZs) | Problem analyzed on a micro scale |
| [29] | slots, security, space | Evaluation of variants using the TOPSIS method | Lack of parameters, lack of data |

(·)—number of analyzed parking lots.

Analyzing the scope of the studies presented in related works, one may notice that they covered a fraction of the data related to parking lots. Many authors base their conclusions on the analysis of a relatively small number of parking lots within one urban agglomeration [15,16,18,20,21,25]. Others, in turn, when analyzing data for larger areas, limit their scope of work, in most cases, to the number of available slots in a given location [19]. Very few researchers draw attention to the fact that the specificity of parking lots might be determined by the type of road infrastructure, its parameters and traffic saturation in a given area. Additionally, the authors of only one of the aforementioned publications included road traffic congestion in the transport network as a factor in their research [15]. There is also a gap in the literature as regards the analysis of the transport network with a specific focus on its saturation with parking lots and the confrontation of research results with applicable international regulations governing the driver's working time. In several works, however, there occurred aspects related to the total area of parking lots [16,19,20,29]. Unfortunately, no analyses that would allow for assessing whether the area is properly developed were presented. Neither did most authors present the research findings that would aim at identifying a link between individual parameters of parking lots and the area in which they occur, as well as between parking lots themselves. Meanwhile, the authors of this paper are of an opinion that a broader perspective on this topic, including a description of the relationship (or lack thereof) between selected parameters, might be a significant

contribution to the described field of knowledge. These considerations might serve as an introduction to further research in the field of transport, logistics, or construction.

The aim of the paper was to analyze the data on parking lots located next to main roads, specifically focusing on their usability for trucks. This data included details such as geographical coordinates, the population density of the area, number of parking slots, elements increasing the security level (lighting, video surveillance, security guards, fences), additional free parking space, and the technical class of the road by which the parking lot is located (road type). For this paper, the following research problems were formulated:

- Are parking lots in large urban areas smaller due to lower land availability, while their saturation in the area is higher?
- What is the distribution of parking lots in the studied transport network, and does it meet the requirements of servicing existing traffic volumes?
- Are parking lots located in areas with low population density equipped with increased security measures against cargo theft?
- Do parking lots located by the roads with higher technical parameters have more parking slots?

Answers to the above-mentioned questions were possible upon the analysis of the data on parking lots, which were previously used to develop a method of allocating parking lots [30].

Considering the above, the authors of this publication, unlike other authors, made a comprehensive analysis of car parks for the actual transport network on a macro scale in the example of Mazovian Voivodeship (Poland). This analysis covered a wide spectrum of parameters, including the often overlooked, but considered important, population density of the area in which there is a car park, as well as technical parameters and the type of the nearby road, which is a novelty in the described area. In addition, an extension of the previous work and an undoubted added value is the analysis of correlations between the individually defined parameters of the car parks, which required a comprehensive process of data collection, processing and validation. For this purpose, the method of Spearman's rank correlation coefficients was used for 113 public and private car parks for trucks located on national roads.

The paper is structured in the following manner. The next section describes theoretical assumptions and the methodology based on the analysis of the parking lot data. Subsequently, the results of the analysis based on the NumPy and Pandas modules from Python technology libraries and Statistica software, based on which relevant graphs and tables, were prepared along with a description of the results. Then, comments on obtained results were provided, conclusions were formulated, and possible further research directions were identified.

## 2. Materials and Methods

### 2.1. Analyzed Area, Data Sources and Research Object

The Mazovian Voivodeship is a centrally located and the largest administrative area of Poland (35,558 km$^2$) which is characterized by one of the largest urbanization indices of nearly 65% [31] and a population density of 153 people per 1 km$^2$ [32]. In comparison to the average EU GDP per capita, estimates for the Mazovian Voivodeship made by Eurostat in 2022 show that the value of this indicator equals 123%. This is to say that the area is comparable to other areas of Western Europe in terms of economic development, including road infrastructure. According to the Statistical Office in Warszawa, in 2020 the length of public hard surface roads in this voivodeship was 39,910 km with a ratio of 112.2 km of such roads per 100 km$^2$ of the total area [33]. In this transport network national roads constitute around 7%. Among these types of transport routes, one might distinguish mainly the A2 motorway and the S2 expressway, which are part of the E30 international route connecting Europe and Asia, as well as the S7 expressways forming part of other European routes (E77 Hungary–Baltic States), S8 (E67 Slovakia–Finland) and S17 (E372 Poland–Ukraine). Based on official reports, the average daily traffic on national roads in 2020 in the Mazovian

Voivodeship amounted to 16.5 thousand vehicles (with average value for Poland being 13.5 thousand), while for international routes it was 35 thousand vehicles (average for Poland being 25.5 thousand) [34]. What is important, the analysis of the vehicle types in the cited source indicates that on some sections of the national road network in the Mazovian Voivodeship, the daily number of heavy goods vehicles passing through significantly exceeded 10 thousand (e.g., on the A2/E30 route), and in many cases, the ration of heavy goods vehicles to the total number of vehicles exceeded 0.1. Such high values recorded at measurement points cause the demand for parking spaces in the transport network of national roads in this area to be comparable to the countries of Western Europe [35].

For the paper, data on 113 parking lots for heavy goods vehicles were collected from different sources. The first of these was the register of parking lots and Travel Service Facilities of the Polish General Directorate for National Roads and Motorways (GDNRM), which is an institution managing road infrastructure, including national roads. These records included both parking lots managed directly by GDNRM (Figure 1, red color), as well as leased and private parking lots (Figure 1, blue color). Another source used to locate parking lots and, among others, measure their area, were satellite images made available courtesy of Google in the Google Maps service. Data on the infrastructure and adjacent parking lots as of 31 December 2020 was supplemented with information obtained from their owners and based on on-site study visits. Data on the population density for car stops were obtained from the database Statistics Poland based on their geographical coordinates [32]. The intensity of road traffic was mapped using the average real travel times along individual roads. The data was collected using the author's webservice created using the Python language, which downloaded the said data in the Distance Matrix API service on Google Maps at two-hour intervals from 25 May 2022 to 22 June 2022.

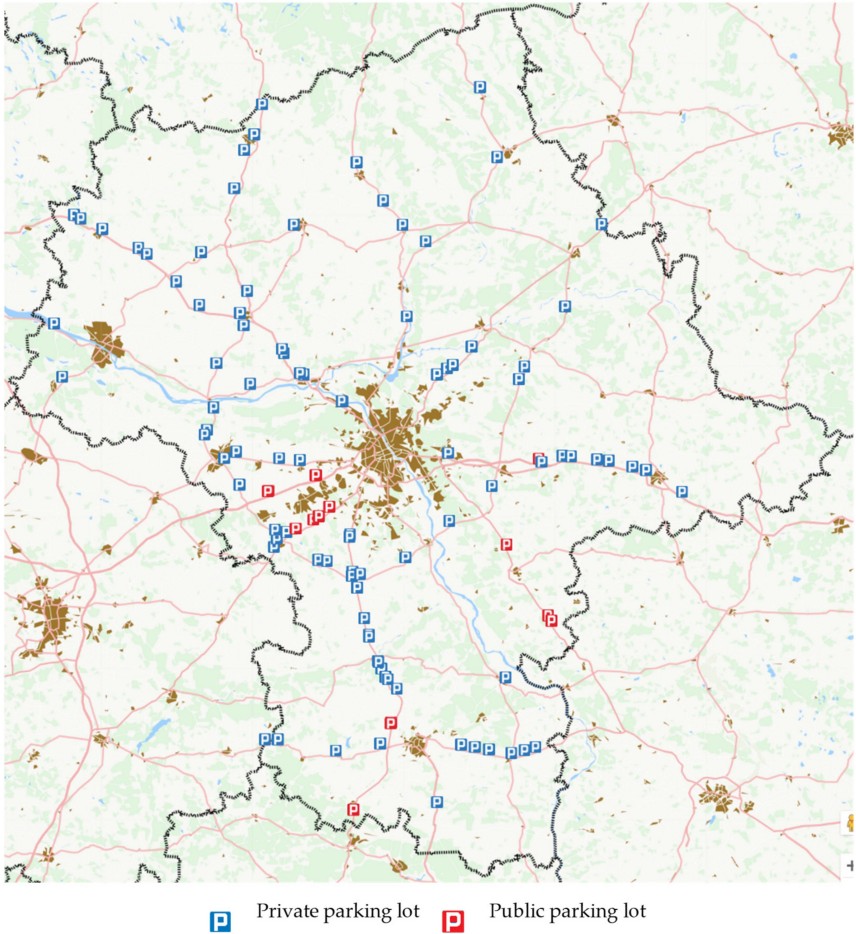

🅿 Private parking lot      🅿 Public parking lot

**Figure 1.** Types and location of the parking lots in analyzed area.

In this study it was assumed that the research objective, a parking lot for trucks, will be marked with a symbol $p$ whereby $p \in P$, where $P$ was specified as a set of parking lots for trucks in the transport network.

### 2.2. General Parameters

The number of parking slots ($p_{num}$) for trucks is one of the most important parameters of parking spaces, affecting its load factor in the time period. The lack of these places is noticeable especially during nighttime and near the main transport routes. Many authors signal the problem of insufficient numbers of parking slots for vehicles transporting cargo, and these effects may interfere with road safety. In numerous studies [19,24,27–29] researchers point to the problem of an insufficient number of parking spaces (including daily parking), which forces truck drivers to stop in prohibited places or increases traffic when searching for optional vehicle stop options. Another study addresses the problem of the unavailability of vehicle stop areas during rush hours, highlighting the negative impact on the transport system and its surroundings. The ratio of demand and supply related to parking spaces necessitates the development of optimization systems and methods, although increasing the number of available parking space is the most dependable solution reducing the likelihood of their unavailability (Table 2).

**Table 2.** Summary of key notations.

| General Notations | Definition |
|---|---|
| $\varepsilon$ | Average number of parking spaces for every 1 km of specific road |
| $p$ | Parking lot for trucks |
| $P$ | Set of parking lots |
| $p_{are}$ | Additional operating space area |
| $p_{bar}$ | Presence of a fence |
| $p_{cctv}$ | Presence of monitoring |
| $p_{den}$ | Population density of the area |
| $p_{grd}$ | Presence of physical protection measures |
| $p_{lat}$ | Longitude |
| $p_{lig}$ | Presence of lighting |
| $p_{lon}$ | Latitude |
| $p_{num}$ | Number of slots allocated for heavy goods vehicles |
| $p_{r\_dis}$ | Chainage |
| $p_{r\_tra}$ | Average weekly traffic intensity level |
| $p_{r\_type}$ | Road type |
| $p_{safe}$ | General security level of the parking space |
| $p_{type}$ | Parking lot type |
| $R$ | Set of roads |
| $r^{gap}$ | Sum of the distances from the boundaries of the analyzed area or the starting point of the indexed road to the nearest vehicle stop areas |
| $r^{num}$ | Indexed road |
| $r_s$ | Spearman correlation |
| $t_{min}$ | Travel time applicable to the analyzed road without traffic |
| $TP$ | Set of types of roads |
| $t_r$ | Average travel time for road with traffic |

Among the parking lots in the national road network, two types of parking lots can usually be distinguished, which are represented by the parameter $p_{type}$: public parking lots ($p_{type}$ = 0), owned by the State Treasury, and private parking lots ($p_{type}$ = 1), usually resulting from conducting additional economic activities. Public parking lots along national roads are primarily intended to provide parking capacity for cars and trucks. Most often, their properties result from the national rules and international guidelines aimed at optimizing parking space and access to the necessary equipment and services. In addition, local authorities and independent research centers publish guidelines aimed at standardizing newly emerging parking spaces [36,37]. On the other hand, the design of parking lots is an issue often raised in the scientific space. For example, Stephan et al. proposed an optimization method using mathematical programming to maximize the number of parking slots that can be reached following a traffic lane [38]. Another issue was raised in the study by Zou and Zhu, who described the problem of using parking space in terms of big data application [39]. However, there are many parking spaces which are not based on

the aforementioned analyses and guidelines, but which play an important role in the road network. These are private parking lots, the existence of which results from the provision of fuel distribution services, catering activities, etc. Yan et al. have noticed the necessity of using even individual private parking slots as part of the sharing economy [40].

It often occurs, that such a parking area functions not as a fully-fledged car stop site, but as a lay-by next to the roadway, and it is a common solution in urban areas. Marshall, Garrick, and Hansen describe this aspect, drawing attention to its usefulness from the perspective of land development and even from the perspective of increasing the level of safety for traffic users [41]. However, there is no agreement among researchers on the final evaluation of this issue [42], and both viewpoints are described by Guo et al. in [43]. Vehicle stops where a non-standard truck can be parked must be considered unconventionally. Often, the specific nature of heavy goods vehicles also includes the issue of problematic dimensions in width and length, which means that the vehicle parking problem should be viewed in a different way from the usually accepted one. As mentioned earlier, parking slots for trucks do not usually exceed 40 m$^2$, and the organization of traffic in the parking lot and the placement of its elements result from these dimensions. Nonetheless, it sometimes happens that a parking lot has a bay in the main lane or an area whose development does not significantly affect other participants (Figure 2). This area contributes to the ease of maneuvering the vehicle, and on the other hand, it can be used for an oversized vehicle, which should be located in such a way as to minimize the need for maneuvers (turning, reversing). This need arises mainly from the transport of long loads. Vehicles with such loads often use an exit ramp and then position themselves extremely close to the left curb (for right-hand traffic) in the parking space without preventing the maneuver of passing and without performing any additional maneuvers. A wide vehicle, on the other hand, does not usually have to use this type of space, as it can, for example, occupy 2–3 truck parking slots. The free area to be used is expressed in the form of a parameter $p_{are}$, which is expressed in m$^2$. This is understood as an area in the shape of a rectangle, which is at least 3 m wide (a value close to the width figure, above which a vehicle should be piloted in most countries of the world) and the maximum possible length resulting from the technical parameters of the vehicle stop area (Figure 2). At the same time, it should be emphasized that the length of this area has a higher priority than its width and that it does not perform important functions at the vehicle stop area and can be temporarily managed for parking purposes. Importantly, the location of the said area must not negatively affect the organization of traffic in the parking lot and should minimize the need to maneuver the vehicle. In this case, the methodology of determining the free space location for an oversized vehicle was based on an expert interview and study by Mikusova and Abdunazarov [20]. The figure below shows an example of an additional space to be used by a vehicle carrying a long load.

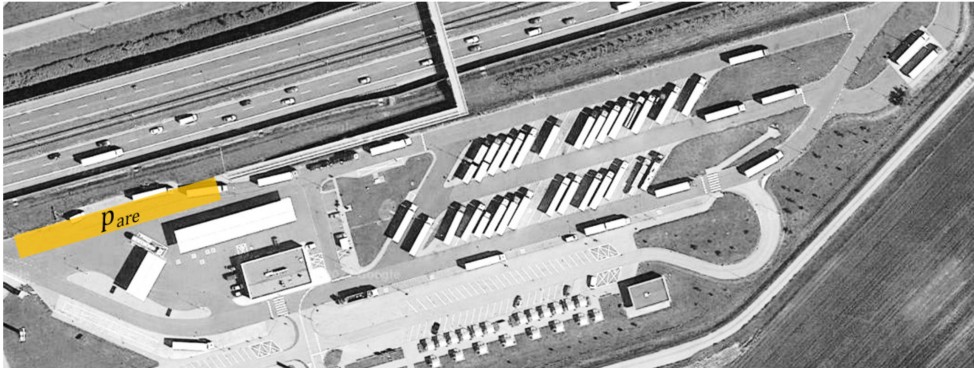

**Figure 2.** The zone possible to arrange as an additional space for trucks ($p_{are}$).

*2.3. Location Parameters*

The location of the parking lots in a space may be considered from the point of view of various criteria. Indisputably, there is an indissoluble link between the occurrence of

vehicle stop spaces and the presence of a transport network, for which they constitute accompanying infrastructure and with which their users interact. For the article, it was assumed that $p_{lat}$ and $p_{lon}$ would be important from the point of view of data visualization in the analyzed area. In turn, the parameters $p_{r\_dis}$ and $p_{den}$ will be further analyzed and compared with other ones to obtain answers to relevant research problems.

For design and then orientation purposes, road sections are parameterized in such a way that both, in the documentation and near the road shoulder, appropriate marking informs what distance separates indicated places from the assumed beginning of the road, indexed with an alphanumeric identifier. Therefore, for the purposes of the article, this convention was used by assigning parking lots a parameter $p_{r\_dis}$ which reflects the location of the parking space along a specific road. Therefore, it will be possible to analyze the distances between parking lots without mapping the structure of the road network in the form of a graph. A comparison of parking opportunities from the point of view of the frequency of available parking spaces and their parking capacities can be made in the following way. Assuming that the vehicle stop areas occurring along some roads can be assigned to the set of these roads' numbers in the transport network, it was assumed that the parameter $\varepsilon$ reflecting the average number of parking spaces for every 1 km of $r^{num}$ road belonging to the set of all the roads $R$ in the analysed transport network distinguishable by its number would be analyzed:

$$\varepsilon = \frac{\sum_{p \in r^{num}} p_{num}}{\sum_{(p,\ p' \in r^{num})} \left| p_{r\_dis} - p'_{rdis} \right| + r^{gap}} \quad \forall p \in P, \ \forall r^{num} \in R \tag{1}$$

where $p_{r\_dis}$, $p'_{rdis}$ are parameters that identify the location of vehicle stop areas occurring consecutively along a particular $r^{num}$ road section and $r^{gap}$ is the sum of the distances from the boundaries of the analysed area or the starting point of the indexed road $r^{num}$ to the nearest vehicle stop areas. In this case, the parameter $\varepsilon$ should proceed towards the minimum, which in the best case would eliminate the need to plan a stopover due to a sufficient number of parking slots occurring along the route. A similar convention was adopted by the Arizona Department of Transportation in the report concerning the analysis of parking opportunities for trucks in the state of Arizona, US [44].

Another issue that was taken into account in the article was the $p_{den}$ population density of the area in which the parking lot is located. It was expressed as the number of people per square kilometre living in a specific district in which particular vehicle stop areas are located. The paper assumes that this parameter will in a sense reflect the actual impact of potential road users on the nearby transport network and will indicate the nature and intensity of the development. In the literature it is a relevant problem, and it is widely commented on. In their work on the example, Hangzhou, Yang, and Huang indicate how a very high level of area urbanization affects the problem of the availability of parking spaces and issues related to parking management [45]. Moreover, these issues are so important that in the literature researchers also focus on the elaboration and development of methods being useful from the point of view of the problem of parking lot availability in the mentioned area, including its availability for heavy goods vehicles [46].

### 2.4. Nearby Road Parameters

For the purposes of the work, a set of types (classes) of $TP$ transport connections were determined at which parking lots may be located in the national road network and the number of which will result from the provisions of relevant legal acts. This set takes the form of: $TP = \left\{ p_{r\_type} : p_{r\_type} 1, \ldots, p'_{r\_type}, \ldots, TP \right\}$ where: $TP$: a set of connection types (classes, where $p_{r\_type} = 1$ for a parking lot located next to a class G road: main road, 2 for GP class: major trunk roads, 3 for S class: expressways, 4 for A class: highways). Distinguishing the parameter $p_{r\_type}$ is also important from the point of view of determining the importance of the road by which the parking lot is located. Consequently, the parameter $p_{r\_type}$ will take the value of 1 and 2 for roads that are not international transport routes,

while 3 and 4 for roads belonging to the European Route. In addition, the need for vehicle stop spaces results primarily from the number of vehicles travelling along a specific section of the road. The number of these vehicles depends on their technical parameters, which include, among others, the number of lanes occurring in one or both directions of traffic.

The phenomenon of congestion is one of the factors influencing the change of a loaded truck's schedule and may generate problems resulting from the need to change the previously planned stopover location [47]. For the purposes of the work, it was assumed that the level of road traffic $p_{r\_tra}$ over the road section adjacent to the vehicle stop area $p$ will be the value from the set of integer numbers {1, ... , 4} reflecting the degree of the road network loading which occurs in the immediate vicinity of the parking lot. The aforementioned weekly load can be calculated using the parameters $p_{r\_tra_l}$, $p_{r\_tra_r}$ reflecting the ratio of the average travel time on a weekly basis (between 7 a.m. and 7 p.m.) to the minimum travel time along the road section occurring immediately 1 km before/after the vehicle stop area:

$$p_{r\_tra} = \frac{p_{r\_tra_l} + p_{r\_tra_r}}{2} \tag{2}$$

The parameters $p_{r\_tra_l}$, $p_{r\_tra_r}$ are distinguishable from each other and the nature of the problem does not force their ranking in relation to the direction of travel. In their case, the following equation should be met:

$$p_{r\_tra_l}, \ p_{r\_tra_r} = \frac{t_r}{t_{min}} \geq 1 \tag{3}$$

both the value $t_r$ and $t_{min}$ can be obtained from web mapping service providers who obtain data from drivers in real time. The data referring to the parameter $t_r$ is the average travel time on the road with traffic. In turn, $t_{min}$ is a travel time applicable to the analyzed road section without traffic. For the purposes of the article, it was assumed that the parameters $p_{r\_tra_l}$ and $p_{r\_tra_r}$ may reflect the intensity of road traffic, however, only if it was ensured that their value could not be affected by other anomalies, such as the occurrence of random road events forcing a temporary limit on the speed for individual sections. Considering the fact that drivers could not comply with the traffic regulations, for all parameter values $p_{r\_tra}$ lower than 1, it was specified that $p_{r\_tra} = 1$. The studies used data collected cyclically every 2 h for a period of 1 week, on the basis of which the parameter values were calculated $p_{r\_tra_l}$, $p_{r\_tra_r}$. For this purpose, the Distance Matrix API (Google Maps) service was used.

### 2.5. Security Parameters

The aspect of the presence of security in the parking lot was raised in the work by Carrese, Mantovani, and Nigro, who assessed, among others, that a proper security system is the most desirable service at vehicle stop areas [24]. The presence of physical protection measures caused the quality of this security to become *good* in the light of the adopted criteria. The parameter reflecting the presence of a security guard was expressed in the form of a binary parameter $p_{grd}$, which in the case of the presence of physical protection measures in the parking space: $p_{grd} = 1$ and $p_{grd} = 0$ in the opposite case.

In the literature, the issue of the presence of monitoring at a vehicle stop area has been analyzed many times, in particular by Welsh and Farrington in terms of crimes aimed at a motor vehicle or its equipment (cargo) [48]. In their work, they stated that monitoring is the most effective in this area, which was later confirmed in other works [24,49]. In connection with this, it was determined that the parameter relating to the presence of monitoring would take the form of a binary parameter $p_{cctv}$. When monitoring is found in a parking lot then $p_{cctv} = 1$, with $p_{cctv} = 0$ in the opposite case.

The aspect of the presence of a fence in a parking space was discussed in the reports [50,51], where it was stated that the presence of barriers surrounding the parking lots is crucial from the point of view of security; see also Carrese, Mantovani, and Nigro [24]. In their work, they stated that the presence of a fence at an illuminated location causes the security of this location to be considered *sufficient*. This parameter in the work took the form of a

binary parameter $p_{cctv}$, such that $p_{cctv} = 1$, when the monitoring is present and $p_{cctv} = 0$ if the parking lot does not have it.

In the literature, attention has been paid many times to the issue of parking lot lighting and the extent to which it affects the safety of drivers and cargo. For example, Rea, Bullough, and Brons analyzed the impact of the type of lighting on visibility at vehicle stop areas and on the sense of security [52]. Their research shows that in most cases unobstructed driving is possible, and people moving around the parking lot are easily visible. Both in the reports [51,53], and in the guidelines describing standards and good practice in the design of parking lots [36], the importance of this aspect is emphasized. However, the presence of the lighting itself, as an element intended to increase the level of security, is insufficient, as noted by, among others, Carrese, Mantovani, and Nigro [24]. The parameter describing the occurrence of lighting at a vehicle stop area is described as $p_{lig}$. When a parking lot is equipped with lighting it is $p_{lig} = 1$, and otherwise $p_{lig} = 0$.

To determine the general level of security at a vehicle stop area $p_{safe}$, the convention described previously in [30] was employed. The parking lots were classified based on their equipment and personnel, where the presence of physical protection measures and monitoring were considered as active protection elements, while lighting and fencing were considered passive protection elements. Depending on their combination of occurrence, parking slots were assigned to 5 categories, with category 1 indicating the absence of security features and category 5 indicating a satisfactory level of security. The method of assigning parking lots to individual categories is presented in the following equations:

$$p_{safe} = 5 \text{ for } \left( p_{cctv} \wedge p_{grd} = 1 \right) \wedge \left( p_{bar} \vee p_{lig} = 1 \right) \tag{4}$$

$$p_{safe} = 4 \text{ for } \left( \left( p_{cctv} \vee p_{grd} = 1 \right) \wedge \left( p_{bar} \vee p_{lig} = 1 \right) \right) \vee \left( \left( p_{cctv} \wedge p_{grd} = 1 \right) \wedge \left( p_{bar} \wedge p_{lig} = 0 \right) \right) \tag{5}$$

$$p_{safe} = 3 \text{ for } \left( \left( p_{cctv} \vee p_{grd} = 1 \right) \wedge \left( p_{bar} \wedge p_{lig} = 0 \right) \right) \vee \left( \left( p_{cctv} \wedge p_{grd} = 0 \right) \wedge \left( p_{bar} \wedge p_{lig} = 1 \right) \right) \tag{6}$$

$$p_{safe} = 2 \text{ for } \left( \left( p_{cctv} \wedge p_{grd} = 0 \right) \wedge \left( p_{bar} \vee p_{lig} = 1 \right) \right) \tag{7}$$

$$p_{safe} = 1 \text{ for } \left( \left( p_{cctv} \wedge p_{grd} = 0 \right) \wedge \left( p_{bar} \wedge p_{lig} = 0 \right) \right) \tag{8}$$

## 3. Results

### 3.1. Parking Parameters Correlations

In order to analyze the correlation of the individual parameters of vehicle stop areas, Spearman's rank correlation was used. For the purposes of this study, it was assumed that the correlation strength at the level $|r_s| < 0.2$ with the significance coefficient $\alpha = 0.05$ would be specified and recognized as worthy of attention. The correlation matrix of individual parameters and the matrix graph of the spread of all variables are presented in Table 3.

**Table 3.** Spearman correlation matrix for all the data.

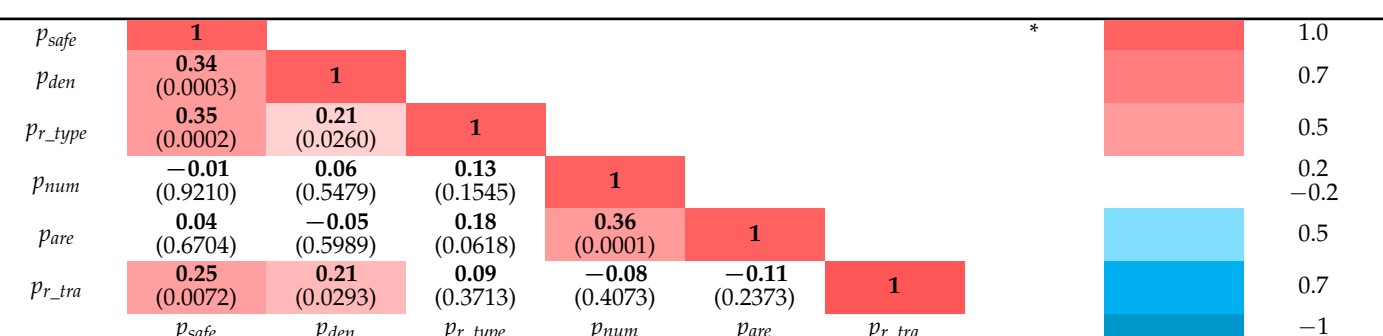

| | $p_{safe}$ | $p_{den}$ | $p_{r\_type}$ | $p_{num}$ | $p_{are}$ | $p_{r\_tra}$ | |
|---|---|---|---|---|---|---|---|
| $p_{safe}$ | **1** | | | | | * | 1.0 |
| $p_{den}$ | **0.34** (0.0003) | **1** | | | | | 0.7 |
| $p_{r\_type}$ | **0.35** (0.0002) | **0.21** (0.0260) | **1** | | | | 0.5 |
| $p_{num}$ | −0.01 (0.9210) | 0.06 (0.5479) | 0.13 (0.1545) | **1** | | | 0.2 / −0.2 |
| $p_{are}$ | 0.04 (0.6704) | −0.05 (0.5989) | 0.18 (0.0618) | **0.36** (0.0001) | **1** | | 0.5 |
| $p_{r\_tra}$ | **0.25** (0.0072) | **0.21** (0.0293) | 0.09 (0.3713) | −0.08 (0.4073) | −0.11 (0.2373) | **1** | 0.7 |
| | $p_{safe}$ | $p_{den}$ | $p_{r\_type}$ | $p_{num}$ | $p_{are}$ | $p_{r\_tra}$ | −1 |

The first value in the cell is correlation strength and the value in parenthesis is the significance coefficient. * Applicable if $\alpha \leq 0.0005$.

Analyzing the results, it was noted that there was no strong correlation among the analyzed parameters. However, among the obtained results, attention was drawn to the correlation $p_{are}$ and $p_{num}$ at the level of 0.36 at $\alpha = 0.0001$, which can be confirmed by the fairly intuitive belief that a larger parking lot is associated with a larger additional operating space. On the other hand, it should be checked whether this space could be adequately developed to increase the parking capacity of a specific parking space. Another clear relationship was the result of the combination of parameters $p_{r\_type}$ and $p_{safe}$ ($r_s = 0.35$ at $\alpha = 0.0002$). It meant that usually parking places located by the road with better technical parameters are characterized by a higher level of security. A clear correlation of the parameters $p_{den}$ and $p_{safe}$ was displayed at a similar level. It demonstrated that parking lots located in an area with a low population density have as a rule lower protection level than the ones in urban areas. This was also confirmed by the correlation $p_{safe}$ and $p_{r\_tra}$ at the level of 0.25 and the significance coefficient equal to 0.0072 as well as $p_{r\_tra}$ and $p_{den}$ at the level of 0.21 and $\alpha = 0.0293$.

The data confirmed the fact that the driving time ratio $p_{r\_tra}$ assumed higher values in the area with a higher population density. Referring to the research problems defined in this work, it is worth mentioning that the analysis did not show a correlation between the number of parking slots and the population density index (Table 4).

**Table 4.** Spearman correlation matrix for parking lots where $p_{r\_type} = 1$ and $p_{r\_type} = 2$.

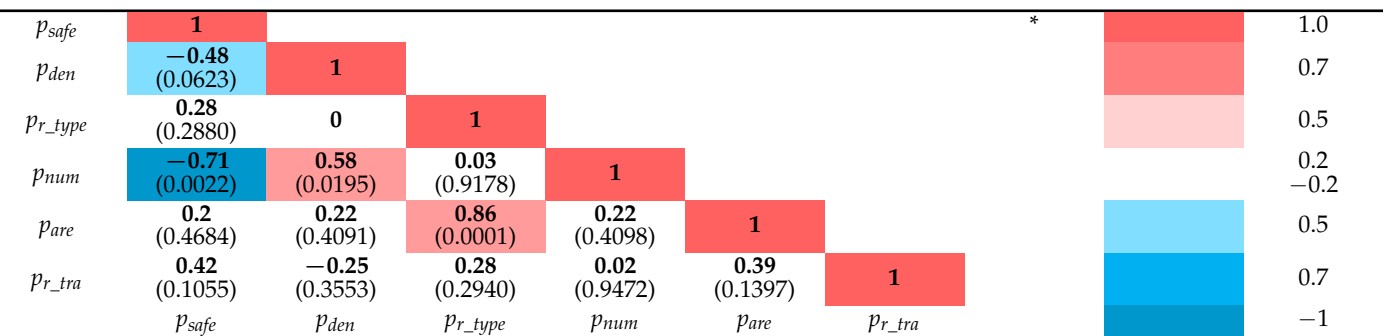

| | $p_{safe}$ | $p_{den}$ | $p_{r\_type}$ | $p_{num}$ | $p_{are}$ | $p_{r\_tra}$ | | |
|---|---|---|---|---|---|---|---|---|
| $p_{safe}$ | 1 | | | | | | * | 1.0 |
| $p_{den}$ | −0.48 (0.0623) | 1 | | | | | | 0.7 |
| $p_{r\_type}$ | 0.28 (0.2880) | 0 | 1 | | | | | 0.5 |
| $p_{num}$ | −0.71 (0.0022) | 0.58 (0.0195) | 0.03 (0.9178) | 1 | | | | 0.2 / −0.2 |
| $p_{are}$ | 0.2 (0.4684) | 0.22 (0.4091) | 0.86 (0.0001) | 0.22 (0.4098) | 1 | | | 0.5 |
| $p_{r\_tra}$ | 0.42 (0.1055) | −0.25 (0.3553) | 0.28 (0.2940) | 0.02 (0.9472) | 0.39 (0.1397) | 1 | | 0.7 / −1 |

The first value in the cell is correlation strength and the value in parenthesis is the significance coefficient. * Applicable if $\alpha \leq 0.0005$.

Next, it was decided to analyze the correlation of parameters based only on vehicle stop slots for which $p_{type} = 0$. The purpose of this analysis was to verify whether the application of Spearman's rank correlation in the case of parking lots built by public institutions as an accompanying infrastructure for the expressway and motorway networks would yield results different from Table 3. The obtained results are presented below in the form of a correlation matrix of individual parameters and a matrix graph of the spread of variables.

Analyzing the data, a clear difference in the strength of the correlation between individual parameters was observed, and even the lack of their occurrence in comparison with the results from Figure 3. The strongest correlation (0.86, $\alpha = 0.0001$) returned was that of $p_{r\_type}$ and $p_{are}$, which in the case of Table 3 was not recorded. This correlation was a confirmation that the planning process and the regulations applicable to the design of vehicle stop areas envisage the connection of the parking space with the road class, and thus its validity in the transport network. The data included roads for which the parameter related to their technical quality $p_{r\_type}$ was 3 and 4. Additionally noteworthy is the negative correlation of parameters $p_{num}$ and $p_{safe}$ at the level of −0.71 at $\alpha = 0.0022$. This demonstrated that the vehicle stop areas possessing fewer parking slots were usually characterized by a higher level of security. The results were obtained for $p_{safe} \in \{3, 4, 5\}$. Another correlation that was not recorded in Table 3 was the relationship $p_{num}$ and $p_{den}$, reaching 0.58 at $\alpha = 0.0195$. This demonstrated that in the case of the designed parking lots in the area with a higher population density, it has an impact on the number of slots

envisaged for heavy goods vehicles. The population density $p_{den}$ was also correlated with the level of parking security $p_{safe}$, because this relationship at the level of $-0.48$ was also considered worth noting, despite a slight deviation from the limit value $\alpha = 0.0623$. The obtained result indicates a completely different relationship than in the case of correlations $r_s = 0.34$ for the entire dataset. It means that vehicle stop areas located in an area with low population density are usually better protected (Figure 4).

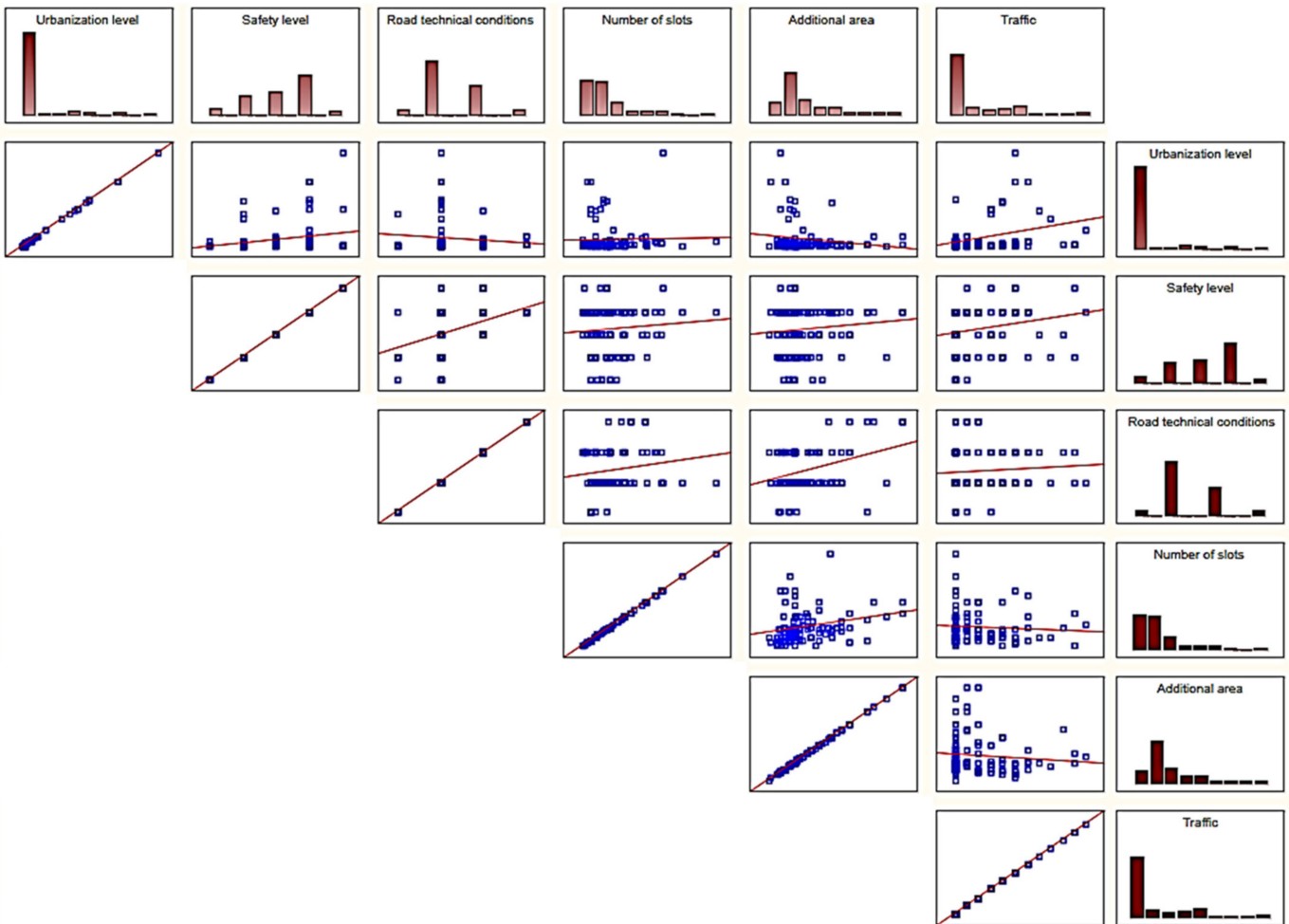

**Figure 3.** Scatter plot matrix for all the data. Urbanization level ($p_{den}$), safety level, ($p_{safe}$), road technical condition ($p_{r\_type}$), number of slots ($p_{num}$), additional area ($p_{are}$), traffic ($p_{r\_tra}$). x-axle and y-axle of each subplot reflect values related to particular types of $p$ parameters.

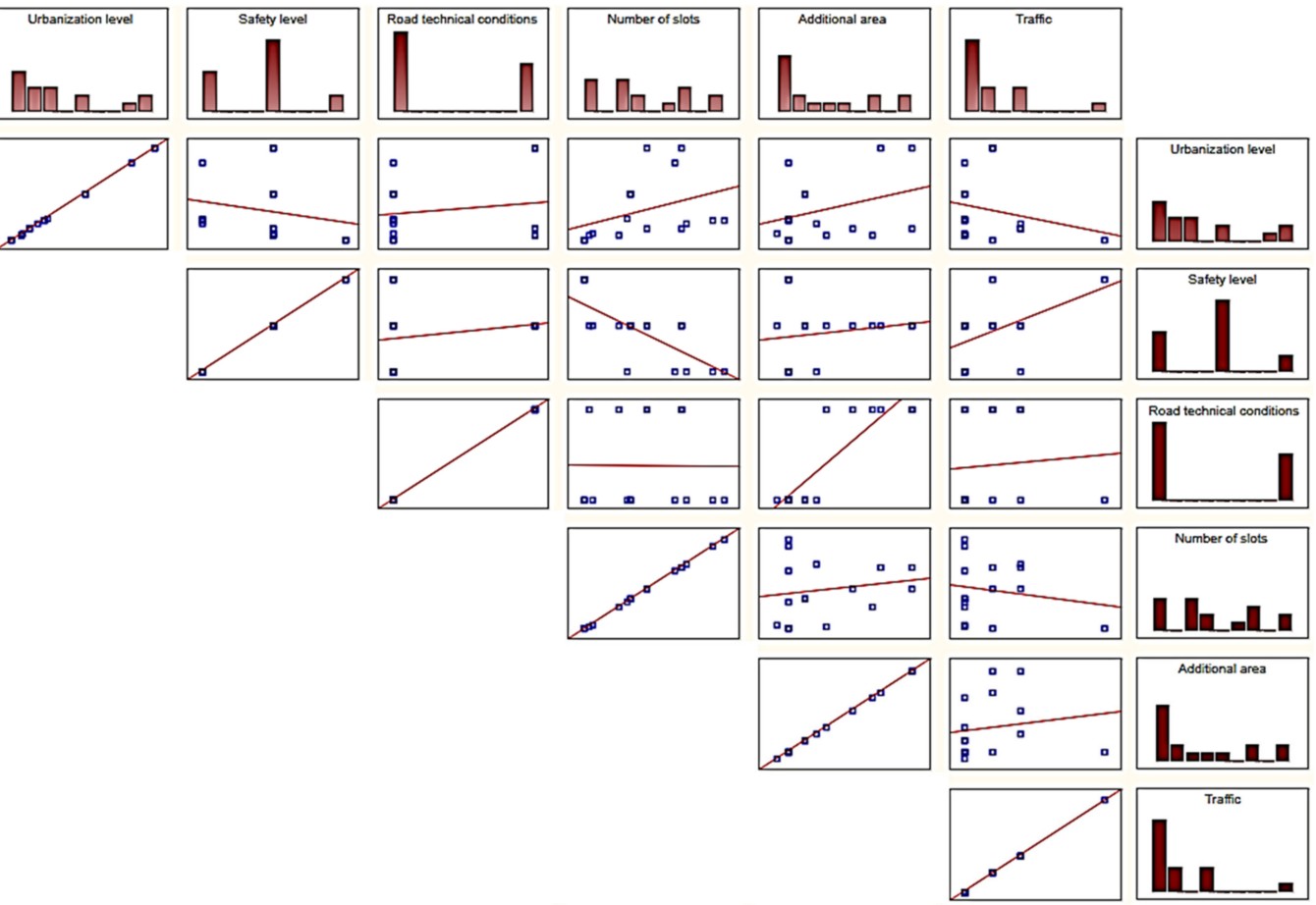

**Figure 4.** Spearman's correlation matrix and scatter plot matrix for parking lots where $p_{r\_type} = 1$ and $p_{r\_type} = 2$. Urbanization level ($p_{den}$), safety level ($p_{safe}$), road technical condition ($p_{r\_type}$), number of slots ($p_{num}$), additional area ($p_{are}$), traffic ($p_{r\_tra}$). x-axle and y-axle of each subplot reflect values related to particular types of *p* parameters.

*3.2. Capacity*

The largest group of parking lots (59%) were vehicle stop areas for which $p_{r\_type} = 2$. The parking lots located by the roads of this class were characterized by a great variety of available parking slots and the largest group of outliers. On the other hand, the interquartile range was relatively small, due to the fact that over 50% of vehicle stop areas in the analyzed group have 10–20 parking slots. The median for this group of parking lots was 17, and the average was 20. The second most numerous group (31%) were the locations for which $p_{r\_type} = 3$. When analyzing Figure 5, particular attention was paid to the relatively large dispersion of data. It was mainly due to the fact that this group of parking lots contained both private parking lots with several slots for trucks, as well as public passenger service facilities with parking lots envisaged for dozens of vehicles of this type. The average for this group of parking lots was 19 with a median of 18. Comparing the obtained results with the data contained in the Arizona Department of Transportation report [49], one can notice a similarity in the parking capacity, as regards public parking lots. The vehicle stops in Arizona were characterized by smaller differences between the number of parking slots, while the parking lots in the Mazovian Voivodeship had a larger capacity for the highest-class roads. The categories of parking lots reaching a similar size (5%) as the locations for which $p_{r\_type} = 3$ were vehicle stop areas located by the lowest class of roads ($p_{r\_type} = 1$) and the highest ($p_{r\_type} = 4$). In the case of the first group, the results presented in Figure 5 indicated a small number of slots envisaged for heavy goods vehicles in all cases and the dispersion of data was negligible. The average number of parking slots was

10 with a median of 8. In the other case, on the other hand, the vehicle stop areas, located next to transportation routes belonging to the motorway network, were characterized by decidedly the largest number of parking slots (average 34 and median 34). It is also worth emphasizing that in the case of this category, all vehicle stop areas were organized by public institutions, which resulted in a relatively small dispersion of data. This may indicate that these places, due to their location, are characterized by similar design requirements.

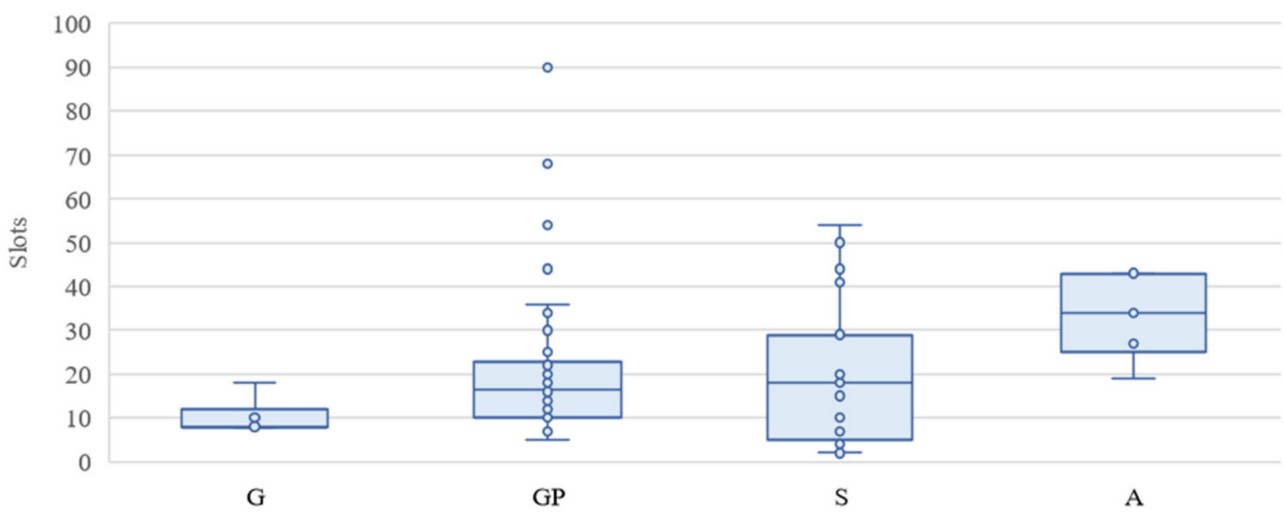

**Figure 5.** Number of parking slots in relation to road technical condition. G ($p_{r\_type}$ = 1), GP ($p_{r\_type}$ = 2), S ($p_{r\_type}$ = 3), A ($p_{r\_type}$ = 4).

For all the collected data, Spearman's rank correlation demonstrated a clear relationship between the number of parking slots and the additional operating space (Table 3), which was not confirmed for the group of parking lots for which $p_{r\_type} \in \{3, 4\}$. The cause can be observed by analysing the graph presented in Figure 6.

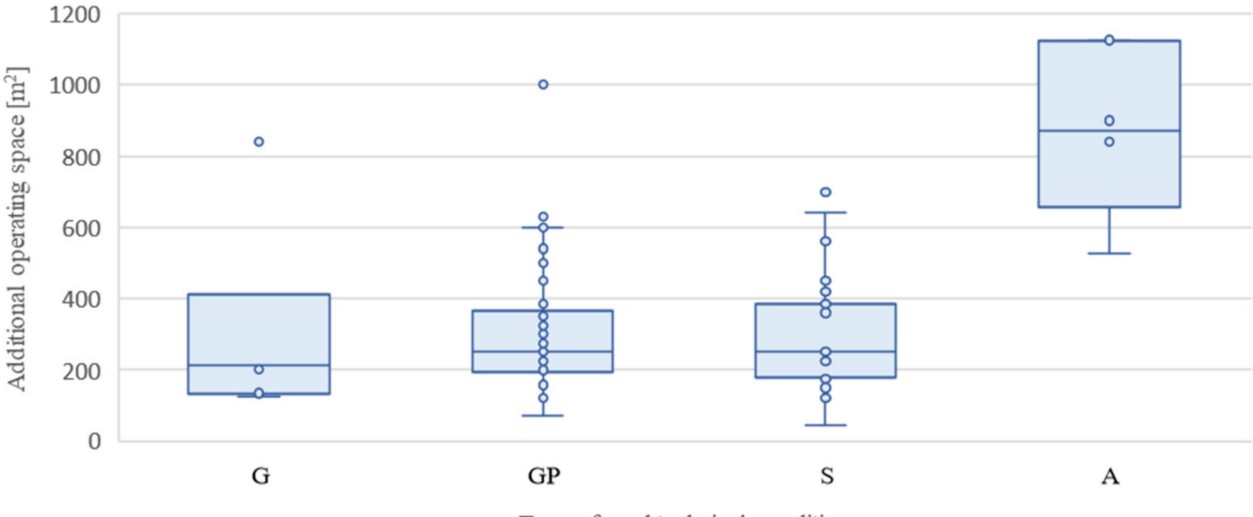

**Figure 6.** The size of parking additional operational space in relation to road technical condition. G ($p_{r\_type}$ = 1), GP ($p_{r\_type}$ = 2), S ($p_{r\_type}$ = 3), A ($p_{r\_type}$ = 4).

For all categories of parking lots located along road sections excluding the motorway network, the interquartile range for additional operating space capable of ensuring vehicle stop space for additional vehicles or oversized vehicle(s) was similar. While the median and

maximum values indicated the aforementioned relationship, the minimum value specific to the parking lots for which $p_{r\_type} = 3$ was lower than in the case of the lower category roads. As in the case of the number of vehicle stop slots for class GP (accelerated traffic main road) and class S (expressway) roads, the dispersion of data was significant, but the differences observed between the results obtained for these categories were relatively small. On the other hand, a clear discrepancy was observable between the additional operating space available for parking lots located next to motorway sections. The average area, in this case, is nearly 900 m$^2$ and was three times larger than the average area of this type for the parking lots for which $p_{r\_type} = 3$. As in the case of the number of parking slots, the category of expressways was characterized by a diversity of results, which can be confirmed by the existence of parking lots having 5 parking slots intended for trucks with 700 m$^2$ of additional operating space and a vehicle stop space for which $p_{num} = 54$ and $p_{are} = 250$ m$^2$.

### 3.3. Location and Security Level

The frequency of parking lots occurrence along the journey route is a special issue in the case of cargo transport and in terms of the driver's working time. In this study, the national roads distinguishable by the number in the Mazovian Voivodeship were analyzed. For each of these roads, the distances between consecutively occurring parking lots were calculated, the results of which are presented in Figure 7.

Analyzing the presented results, one can notice very clear discrepancies between both the number of parking slots along particular roads, as well as the distances between the consecutive parking lots. The roads characterized by the shortest distances between consecutive parking lots are national roads being locally important, i.e., 12. On the other hand, for some of the roads, it was possible to distinguish only one parking place along their route in the analyzed area. Such roads included routes 9 and 53, which both achieved satisfactory values at the level of approx. 20 km. Slightly worse results were obtained by the S17b road (southbound), while the only parking lot on the road number 48 was located nearly 100 km from the border of the voivodeship. In such situation, there is a significant risk resulting from the probability of exceeding the driver's maximum driving time.

The longest distance between the nearest parking lot and the border of the voivodeship was recorded for the national road 62 and amounted to over 160 km. In the case of other international routes, the results can be assessed as satisfactory, as the distances presented in Figure 7 generally, did not exceed 60 km. Taking into account both the average distance between parking lots as well as their number, a highly favorable situation in terms of parking capacity occurred in the case of the roads 12, 50, and 10, which resulted from a large number of private parking lots located along them, and along the international route E77. When comparing these results with the data from the region, it turns out that they are similar. According to Gnap and Kubíková, the recommended distance between parking lots classified as large ones (A–C) should be between 30 and 200 km [31]. In most cases, these assumptions are fulfilled (similarly in the Slovak Republic, as discussed in the aforementioned source).

However, the analysis of the data above did not include the parking capacities of individual vehicle stop areas. For this purpose, it was necessary to refer to the previously defined parameter $\varepsilon$. In the graph in Figure 8, the ratio of the number of all parking slots located along the road $r^{num}$ to the length of this road in the analysed area is presented.

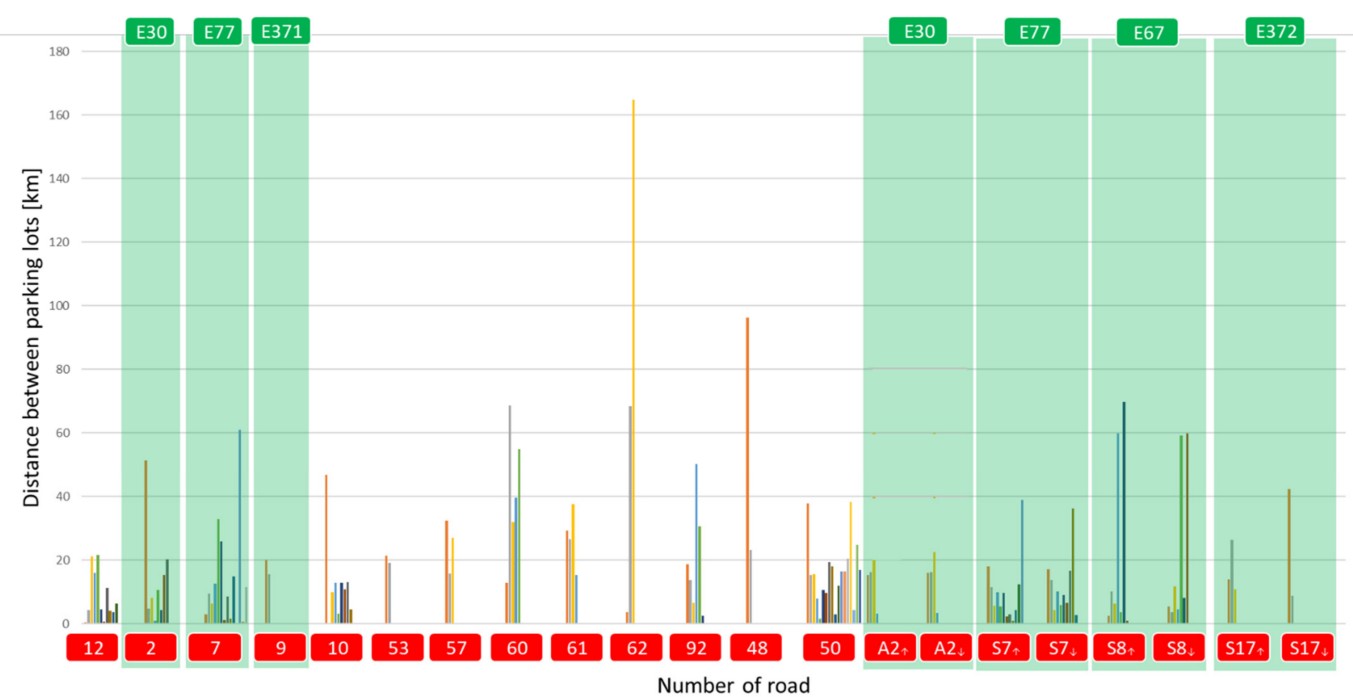

**Figure 7.** Distance between successive parking lots along particular roads (green color with labels means roads from inter-European road system, ↑↓ reflect distinguishable directions of the highways and expressways).

This ratio was expressed as the number of parking slots per 1 km of the road. Analyzing Figure 8, a clear disproportion between parking lots in terms of the analyzed parameter is noticeable. The highest value ($\varepsilon = 1.82$) was obtained for road 50, and slightly lower for roads 12 ($\varepsilon = 1.68$) and 10 ($\varepsilon = 1.58$), despite a significant difference in the number of parking lots, which suggested differences in the average number of parking slots for these roads. Analysing the results for international transport routes, the most favourable situation occurred for the roads S8/E67 and S7 together with the road 7 (E77) conjoining it, while a slightly worse situation occurred in the case of the E67 (A2/2) route. However, discrepancies with regard to the parameter $\varepsilon$ could be observed depending on the driving direction. In the case of the S8/E67 route, the discrepancy was 0.54 and was greater than $\varepsilon$ obtained for 33% of the roads. Similar results were obtained for the E67 route, where both the roads 2 and A2 obtained results at the level of $\varepsilon \cong 0.8$. Different results were obtained in the case of the S17 road, where depending on the direction of travel the number of parking slots per 1 km of the road was 0.35 and 0.92. The aforementioned disproportion was influenced by the fact that in the case of this route there was a difference in the number of parking lots in the analyzed area.

Spearman's rank correlations for $p_{den}$ and $p_{safe}$, depending on the type of parking lots, indicated a relationship between these parameters, although its specificity varied. For the analysed data, 73% of parking lots were located in an area where population density did not exceed 100 people per 1 km². On the other hand, only 10% of parking lots were located in an area characterized by a very high population density (over 1000 people per 1 km²). Analyzing the distribution of parking lots in the transport network (Figure 9), it was possible to observe a high density of parking lots within urban areas. This was especially visible in the case of the Warsaw Agglomeration. The distribution of parking lots from the point of view of the route and location in the analyzed area was also particularly noticeable. In Figure 9, there is a clear discrepancy between the number of parking lots in the northern and southern parts, as well as in the western and eastern parts. The northern and north-eastern regions stood out the most in terms of the lack of parking places, and it was significant inasmuch as the E67 international route, among others, runs across its

area. In terms of the size of parking lots, the central part of the voivodeship was definitely distinguished, especially in locations where international routes were intersected.

From the point of view of the safety of vehicle stop areas, the best equipment appeared in the parking lots located near international routes and the parking lots located in an area characterized by increased population density. It was visible, as in the previous case, on the example of the Warsaw Agglomeration, where an accumulation of $p_{safe} = 4$ and $p_{safe} = 3$ parking lots occurred, while the northern and north-eastern part was dominated by parking lots with $p_{safe} = 1$. It is also worth noting that in the case of the largest vehicle stop areas, the dominant security parameter reached $p_{safe} = 3$ and higher. The parking lots characterized by the highest level of security ($p_{safe} = 5$) were located either in an area with a high population density along transportation routes with $p_{r\_type} = 2$ or a very low population density along the roads for which $p_{r\_type} = 3$. Their total share in all vehicle stop areas was estimated at 4%. More than 27% of parking lots could be assessed as insufficiently equipped with elements affecting the level of protection ($p_{safe} = 1$ and $p_{safe} = 2$). In their case, the parameter $p_{r\_type} = 1$ and $p_{r\_type} = 2$. Considering the totality of the security results, it can be concluded that they are satisfactory, taking into account the results obtained by Carrese et al. in [26]. In their study on the level of safety in the Lazio region (which can be compared with the Mazovian Voivodeship due to the presence of agglomerations of European importance), truck drivers estimated that only 7% of vehicle stop areas were fenced, 13% had an alarm system, 27% had a monitoring system, and in 55% of cases lighting conditions were poor or insufficient.

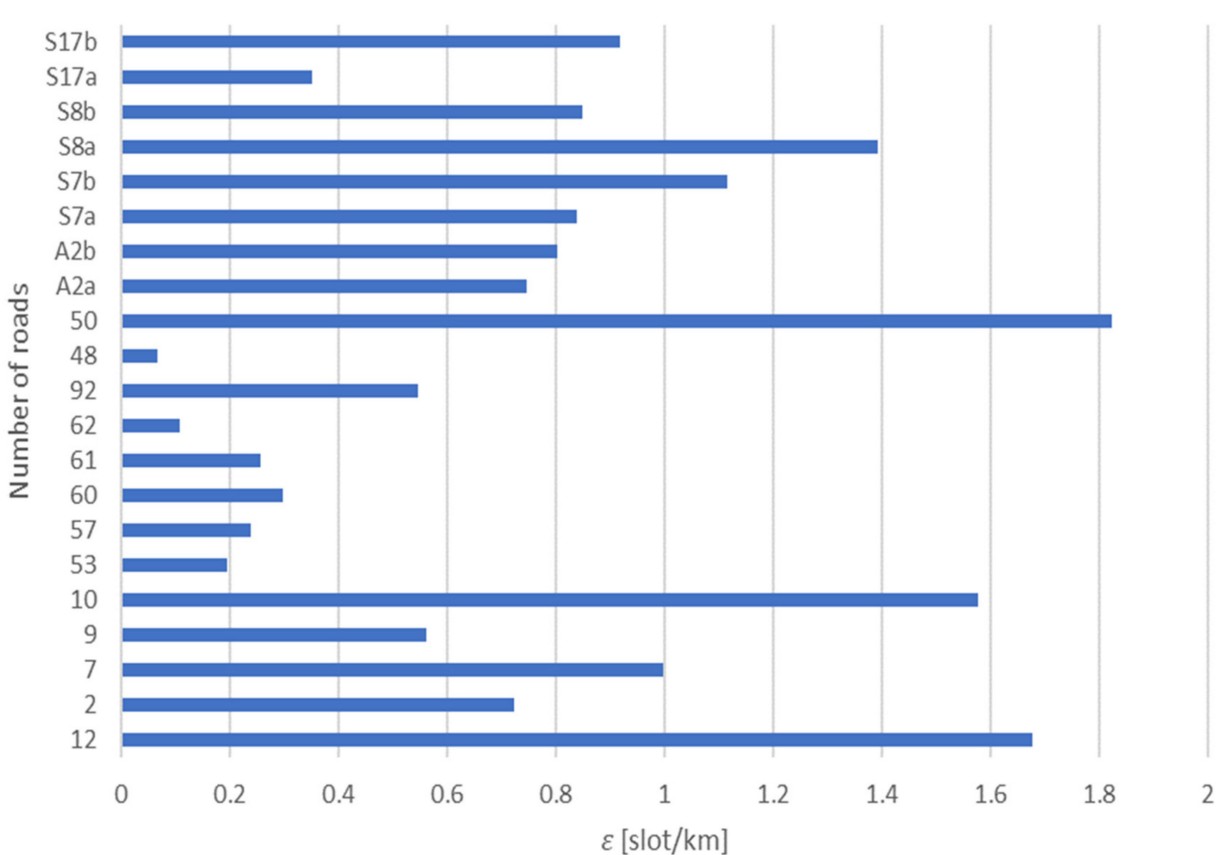

**Figure 8.** Value of $\varepsilon$ parameter in relation to particular roads.

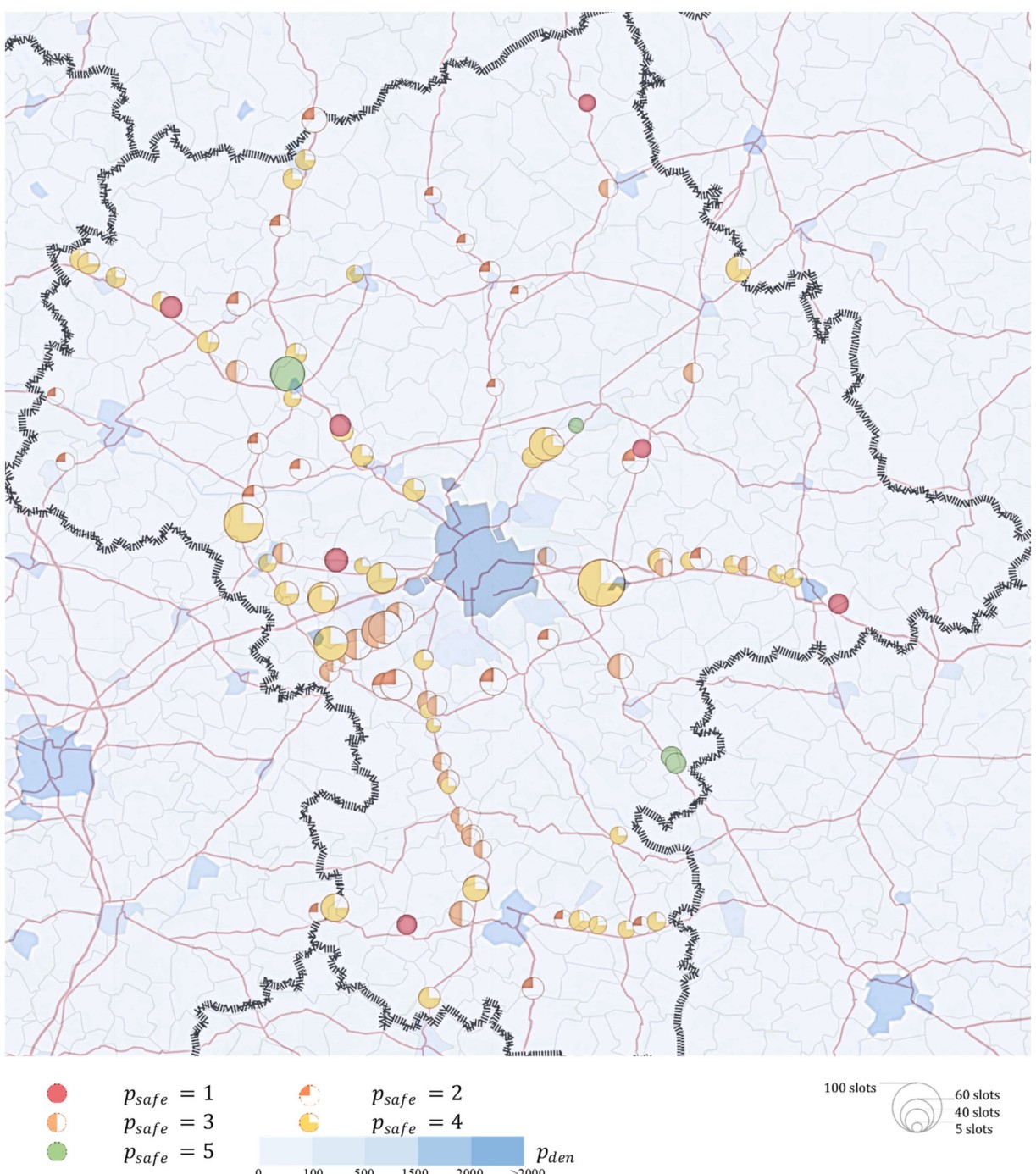

**Figure 9.** Location, number of slots, and security level for parking lots in the Mazovian Voivodeship.

## 4. Discussion

In this section, the findings obtained from the analysis of parking spaces are discussed and compared with the results presented by other researchers. In addition, the limitations of the adopted methodology were identified.

Due to the numerous limitations of the other researches studies and the small amount of analyzed data, comparing the findings with the works presented in Table 1 was problematic. Nevertheless, several works can be distinguished, based on which it was possible to compare the results obtained. In the work of Fleger et al. the average number of parking spaces in a public rest area within the national highway system (NHS) in the US was 18 [23]. Comparable results were obtained for GP (20) and S (19) class roads. In the case of private

parking lots in the USA, this indicator was in the range of 50–84, which corresponds to the maximum values obtained in this study. Considering the fact that average values were used, parking possibilities in urbanized states at a similar level as Mazovian Voivodeship are incomparably greater. In the study conducted by Gnap and Kubíková [27] a comparable saturation with parking lots of the analyzed section of the D1 motorway (every 10–20 km) to the A1 motorway in the study (approx. every 20 km) was obtained, although the authors presented a clearly lower average of parking slots per single parking lot compared to our results, 24 vs. 34.

For aspects of parking security measures, the results obtained are much more satisfactory than those described by Carrese et al. for the Lazio Region [24]. In their study, drivers estimated that the monitoring system was present in 27% of cases, the fence in 7%, and the parking lots were not lit enough in 22% of cases. In this study, 47%, 50%, and 13%, respectively, were obtained in these categories.

Analyzing the results presented by Arizona Department of Transportation [49] regarding the number of parking slots per 1 km of the road, it may be concluded that the parking capacity for trucks in the Mazovian Voivodeship in terms of public parking lots is definitely greater than in the state of Arizona. The average value of the parameter $\varepsilon$ in this area was about 0.55, while for most roads where only public parking lots are found (A2/S17) the obtained value $\varepsilon$ was definitely higher. The only exception was one of the roadways of the E372 route, for which $\varepsilon = 0.35$.

A significant limitation of this study was the lack of data regarding the occupancy of parking lots in the relevant time period, which is an issue widely commented on in the literature. Thus, this work does not answer the question of whether the analyzed parking lots meet the needs resulting from the demand for a parking space. On the other hand, this demand is variable and its analysis may give different results depending on many various parameters. For this reason and due to technological limitations (lack of current occupancy monitoring systems), the analysis of parameters invariable in time was adopted.

Another problematic aspect is the formulated parameter of traffic. In the adopted methodology, it was reflected by the average travel time, which sometimes may not be related to the phenomenon of congestion. Moreover, this parameter reflected the average weekly travel time and was therefore reduced due to night hours. As a result, it makes deeper analysis difficult due to small differences between individual parameters $p_{r\_tra}$.

## 5. Summary and Conclusions

The Mazovian Voivodeship, characterized by similar parameters to Western Europe countries in terms of population density, per capita GDP level, and in terms of the expansion of the transport network, exhibited a relatively large variety of vehicle stop areas within its borders. Based on the results obtained using the Spearman's rank correlation coefficient, this diversity demonstrated a connection with the quality of the nearby linear infrastructure, the level of population density, and the level of the transport network loading. In the case of vehicle stop areas which were designed according to specific guidelines, a special link is noticeable between the road type and the dimensions of the vehicle stop area (0.86, $\alpha = 0.0001$), thus consequently affecting the available parking space. Taking into account all the data, the analysis did not demonstrate a clear connection between the number of available parking slots and the population density of a given area (0.06, $\alpha = 0.5479$), although in the case of designed parking lots, this correlation was significant (0.58, $\alpha = 0.0195$). This may indicate that this parameter is taken into account in the planning process. On the other hand, there is a significant density of parking places in areas adjacent to highly urbanized terrain, while national roads are usually lacking in parking slots for trucks in urban areas. This fact raises the necessity to take into account vehicle stop areas in the process of planning a journey route. The lack of this element may result in the need to go off the main route in an urban area in order to search for a parking lot, creating thus difficulties in maneuvering the vehicle, and even lead to entering areas with restrictions on truck traffic.

The distribution of parking lots in the transport network can be assessed as uneven. In the case of national roads with low population density, which do not serve as international transport routes, the distance between consecutive parking lots for heavy goods vehicles may be up to 100 km and more. On the other hand, national roads, which may be an alternative to the main routes due to the circumvention of highly urbanized areas, are characterized by a high density of vehicle stop areas. Importantly, there is a disproportion between parking opportunities for the main international (dual carriageway) routes from the point of view of the direction of travel. The parameter $\varepsilon$, according to which the aforementioned parking possibilities were assessed, demonstrated that the vehicle stop areas located next to national roads of higher classes usually feature the highest index of parking slots number per 1 km of the road ($\varepsilon > 1$). On the other hand, the results indicated that there are national roads with $p_{r\_type} = 2$, for which the parameter value $\varepsilon$ was significantly higher than in the case of the highest-class roads (e.g., $\varepsilon = 1.82$ for route no. 50). Moreover, motorway sections did not show the highest values in this field, and even oscillated around the average ($\varepsilon \approx 0.8$).

In the case of parameters describing the safety of vehicle stop areas, studies showed that for the totality of data there exists a connection between the population density of an area and the level of security in parking spaces (0.34, $\alpha = 0.0003$). In general, the higher the level of urbanization of an area, the higher the level of parking lot security, which in a way contradicts the expected results. On the other hand, it turns out that in the case of designed parking lots, the level of parking lot security increases with the decrease in population density and this is a clear correlation ($-0.48$, $\alpha = 0.0623$).

Summing up the research results, the authors note a clear need to supplement the future studies with parameters related to the actual demand for parking slots, which is possible to model based on the parameter $p_{r\_tra}$ and systems monitoring the occupancy of individual parking lots in real time. Taking into account the traffic load of the national roads reported by relevant public institutions, it is estimated that the vehicle parking potential needs to be increased in the north-eastern area of the voivodeship along the E67 route and alternative routes. According to the authors, future research should also include proposals for optimizing the method of rational use of space for parking infrastructure, along with a description of innovations related to the parking organization and management. In order to analyze the legitimacy of building additional parking lots in any area, the approach presented in this article can be used.

**Author Contributions:** Conceptualization, I.B. and E.M.; methodology, I.B.; software, I.B.; validation, I.B.; formal analysis, I.B.; investigation, I.B.; resources, I.B.; data curation, I.B.; writing—original draft preparation, I.B.; writing—review and editing, I.B.; visualization, I.B.; supervision, E.M.; project administration, I.B.; funding acquisition, I.B. All authors have read and agreed to the published version of the manuscript.

**Funding:** This research has been partially founded by project granted from National Science Centre, Poland (NCN-MINIATURA-5 DEC-2021/05/X/ST8/01669).

**Institutional Review Board Statement:** Not applicable.

**Informed Consent Statement:** Not applicable.

**Data Availability Statement:** Not applicable.

**Acknowledgments:** This publication was funded by the Military University of Technology, grant number UGB No. 752. This support is gratefully acknowledged.

**Conflicts of Interest:** The authors declare no conflict of interest.

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
