# Peer review of "Characteristics of Parking Lots Located along Main Roads in Terms of Cargo Truck Requirements: A Case Study of Poland"

_sustainability, doi:10.3390/su142315720_

Round 1
Reviewer 1 Report
A summary:
This manuscript explores the characteristics of parking lots next to the national roads in Mazovian Voivodeship, Poland, and the correlation of multiple parameters in individual parking lots and those related to nearby roads. I think that the authors did a good job in analyzing the parameters of parking lots. The results in this manuscript involve a lot of interesting work and should be shared with our scientific community to enhance our understanding of parking issues and to provide a guide to transport infrastructure and parking design. The following are my detailed comments and I hope that those would be helpful to improve the quality of this manuscript.
In Tables 3 and 4, and Figures 3 and 4, the numbers and subplots are exactly the same along the diagonal. I recommend authors demonstrate either half to avoid distracting readers’ attention.
Specific comments:
L17: “capacity of parkings” -> the capacity of parking
L64: “The financial aspects… was…” -> The financial aspects…were…
L65: “basing on” -> based on
L99: “have” -> has
L138: “with specific focus” -> with a specific focus
L172: “of total area” -> of the total area
L276: What does “CCTV” stand for?
L406: “in following” –> in the following
L415: Can you explain what the numbers mean in and out of the parenthesis?
L433: Can you specify what are the x- and y-axis in the subplots?
L439: This table should be Table 4.
L550: Figure 7 is a little confusing. Can you provide more explanation about it? For example, what does the green shadow stand for? Is there a color bar missing?
L627: The symbol for security level is misleading. You can use either color or percentage of the pie to represent the security level. It is confusing to use them together.
Author Response
Dear Sir/Madame,
Please find the upgraded manuscript with the following improvements:
- The manuscript was translated by a professional translation office in accordance with ISO standards. The text was reviewed once again and mentioned mistakes were corrected;
- CCTV was replaced by a "monitoring system";
- The convention of the presented results was described below the Tables 3 and 4;
- Duplications of the results in Tables 3, 4 and figures 3, 4 have been removed;
- We added the additional description below figures 3, 4 regarding the x and y axis;
- We added the additional description below the figure 7;
- Changes in the figure 9 were impossible due to no access to the script which generated the marks. It was based on Google Maps engine. We would kindly ask you to give us the opportunity to save this figure without any changes.
- We included numerical results in the abstract;
- We rebuilt the introduction and deleted the studies descriptions of other researchers. Instead of that we created a new table with parameters and methodology of related works and focused also on the weaknesses of their studies.
- We highlighted the aim of the paper strongly;
- We put emphasize on the novelty of the paper at the end of the introduction section.
- We added the discussion section, where we compare our findings with the results of different authors;
- The limitations of our studies were highlighted at the end of the discussion section;
- We added numerical results to the summary and conclusions section;
- We developed proposals for future researches at the end of the paper.
We would like to thank you for all of necessary remarks, which gave us the different point of view on our study. We believe that after completed adjustments the paper will fulfill requirements to being published in MDPI.
Regards,
Capt. Igor Betkier, PhD

Reviewer 2 Report
This manuscript is within the scope of the journal. The topic is interesting. However, some problems still need to be clarified and revised. I would recommend for a major revision of the manuscript after addressing the significant improvement requirements.
The point of concern are as follows:
1. The introduction is disordered and aimless. Author/ Authors just simply list the lows of literature reviews. However, the important information, such as innovations, focused issues, methods, and value of this article, are missing or oversimplified. It is necessary to rewrite the introduction. The literature review has been how written which leads to misunderstanding.
2. The novelty of the work must be clearly addressed and discussed, compare your research with existing research findings, and highlight novelty, (compare your work with existing research findings and highlight novelty).
3. The main objective of the work must be written more clearly. Also, you should have a subsection on the strengths and limitations of your study.
4. Nomenclature section is also missing. Add completely in the table.
5. There is a strong need to improve the manuscript's language as it is inappropriate for the Journal of Sustainability. The text has lots of grammatical errors. Most of the sections cannot be read well.
6. There are some typos in the text, refine them. According to the MDPI Sustainability journal template, edit the text completely.
7. Author/ Authors need to add more results in the conclusion section and abstract section (especially numerical results) to thoroughly support the main findings. The results are not clear and complete and must be more. The conclusion section should be rewritten with the complete data.
8. It is also suggested to include a paragraph at the end of the conclusion section that should describe the applications of the present work, recommendations, and future scope to other researchers.
Author Response
Dear Sir/Madame,
Please find the upgraded manuscript with the following improvements:
- The manuscript was translated by a professional translation office in accordance with ISO standards. The text was reviewed once again and mentioned mistakes were corrected;
- We added a summary of key notations table;
- We included numerical results in the abstract;
- We rebuilt the introduction and deleted the studies descriptions of other researchers. Instead of that we created a new table with parameters and methodology of related works and focused also on the weaknesses of their studies.
- We highlighted the aim of the paper strongly;
- We put emphasize on the novelty of the paper at the end of the introduction section.
- We added the discussion section, where we compare our findings with the results of different authors;
- The limitations of our studies were highlighted at the end of the discussion section;
- We added numerical results to the summary and conclusions section;
- We developed our proposals for future researches at the end of the paper.
We would like to thank you for all of necessary remarks, which gave us the different point of view on our study. We believe that after completed adjustments the paper will fulfill requirements to being published in MDPI.
Regards,
Capt. Igor Betkier, PhD

Round 2
Reviewer 2 Report
I would recommend accepting the manuscript.
The authors have addressed the required changes in the revision and all comments are addressed properly.